# Dietary Supplement Intake and Fecundability in a Singapore Preconception Cohort Study

**DOI:** 10.3390/nu14235110

**Published:** 2022-12-01

**Authors:** Chee Wai Ku, Chee Onn Ku, Liza Pui Chin Tay, Hui Kun Xing, Yin Bun Cheung, Keith M. Godfrey, Marjorelee T. Colega, Cherlyen Teo, Karen Mei Ling Tan, Yap-Seng Chong, Lynette Pei-Chi Shek, Kok Hian Tan, Shiao-Yng Chan, Shan Xuan Lim, Mary Foong-Fong Chong, Fabian Yap, Jerry Kok Yen Chan, See Ling Loy

**Affiliations:** 1Department of Reproductive Medicine, KK Women’s and Children’s Hospital, Singapore 229899, Singapore; 2Duke-NUS Medical School, Singapore 169857, Singapore; 3Faculty of Science, National University of Singapore, Singapore 117546, Singapore; 4Department of Obstetrics and Gynaecology, KK Women’s and Children’s Hospital, Singapore 229899, Singapore; 5Yong Loo Lin School of Medicine, National University of Singapore, Singapore 117597, Singapore; 6Program in Health Services & Systems Research and Center for Quantitative Medicine, Duke-NUS Medical School, Singapore 169857, Singapore; 7Tampere Center for Child, Adolescent and Maternal Health Research, Tampere University, 33014 Tampere, Finland; 8Medical Research Council Lifecourse Epidemiology Centre, University of Southampton, Southampton SO16 6YD, UK; 9National Institute for Health Research Southampton Biomedical Research Centre, University of Southampton and University Hospital Southampton National Health Service Foundation Trust, Southampton SO16 6YD, UK; 10Singapore Institute for Clinical Sciences, Agency for Science, Technology and Research (A*STAR), Singapore 117609, Singapore; 11Yong Loo Lin School of Medicine, National University of Singapore, National University Health System, Singapore 119228, Singapore; 12Department of Paediatrics, Yong Loo Lin School of Medicine, National University of Singapore, National University Health System, Singapore 119228, Singapore; 13Khoo Teck Puat-National University Children’s Medical Institute, National University Hospital, National University Health System, Singapore 119074, Singapore; 14Department of Maternal Fetal Medicine, KK Women’s and Children’s Hospital, Singapore 229899, Singapore; 15Department of Obstetrics & Gynaecology, Yong Loo Lin School of Medicine, National University of Singapore, National University Health System, Singapore 119228, Singapore; 16Saw Swee Hock School of Public Health, National University of Singapore, National University Health System, Singapore 117549, Singapore; 17Department of Paediatrics, KK Women’s and Children’s Hospital, Singapore 229899, Singapore; 18Lee Kong Chian School of Medicine, Nanyang Technological University, Singapore 636921, Singapore

**Keywords:** supplements, fertility, preconception, time to pregnancy, folic acid, evening primrose oil

## Abstract

Subfertility is a global problem affecting millions worldwide, with declining total fertility rates. Preconception dietary supplementation may improve fecundability, but the magnitude of impact remains unclear. This prospective cohort study aimed to examine the association of preconception micronutrient supplements with fecundability, measured by time to pregnancy (TTP). The study was conducted at KK Women’s and Children’s Hospital, Singapore, between February 2015 and October 2017, on 908 women aged 18-45 years old, who were trying to conceive and were enrolled in the Singapore PREconception Study of long-Term maternal and child Outcomes (S-PRESTO). Baseline sociodemographic characteristics and supplement intake were collected through face-to-face interviews. The fecundability ratio (FR) was estimated using discrete-time proportional hazard modelling. Adjusting for potentially confounding variables, folic acid (FA) (FR 1.26, 95% confidence interval 1.03–1.56) and iodine (1.28, 1.00–1.65) supplement users had higher fecundability compared to non-users. Conversely, evening primrose oil supplement users had lower fecundability (0.56, 0.31–0.99) than non-users. In this study, preconception FA and iodine supplementation were associated with shortened TTP, while evening primrose oil use was associated with longer TTP. Nonetheless, the association between supplement use and the magnitude of fecundability changes will need to be further confirmed with well-designed randomised controlled trials.

## 1. Introduction

Subfertility is defined as any form of reduced fertility with a prolonged time to pregnancy (TTP) [1]. With an approximately two-fold decrease in the global fertility rate of 4.7 children per woman in 1950 to 2.4 in 2021 [2], the impact of low fertility is a serious public health concern. At a personal level, subfertility adversely affects the lives of couples struggling to conceive, resulting in an increased risk of pervasive guilt, dissatisfaction with themselves and their marriage, psychosexual problems, and economic hardship to finance expensive and invasive infertility treatments [3,4,5,6]. It is therefore imperative to investigate possible solutions that are effective and easily accessible for couples who wish to have children.

Female fertility, defined as reproductive performance culminating in a live birth [7], is influenced by many factors. Medical conditions aside, lifestyle factors such as body mass index (BMI), smoking exposure, alcohol consumption and dietary intake affect the chances of pregnancy [8,9,10]. Preconception micronutrient supplementation is an increasingly common practice, with 63% and 71% of Caucasian and Asian women in Australia and Singapore taking supplements, respectively [11,12,13]. Dietary micronutrient supplements, which can be taken in the form of single or multivitamins and minerals [14], have been suggested to have possible effects on fertility [9,15,16]. Micronutrients, including folic acid (FA), vitamins B6, B12, C, D, E, iron, zinc, selenium, iodine, fish oil and phytoestrogen, were found to play a role in female fertility as shown in in vivo and clinical studies that mainly conducted in Western populations [9,15,17,18,19]. To date, little is known about the impact of evening primrose oil, a form of phytoestrogen, on pregnancy rates, even though it is commonly featured on fertility websites and taken by women trying to conceive [20,21,22]. Despite the controversy surrounding phytoestrogens and fertility [9,23,24], there is a paucity of published data on the effects of evening primrose oil on fertility. Furthermore, existing supplement studies have focused on the pregnancy rate as an outcome, without accounting for the TTP [15,16,25]. Fecundability, measured by TTP, is defined as the physiological potential of pregnancy in a menstrual cycle. To our knowledge, only three studies, specific for FA, fish oil, and iron, have investigated the use of these preconception supplements on fecundability in Western setting [26,27,28]. FA supplementation, either on its own or in a multivitamin, showed a positive association with fecundability, with a fecundability ratio (FR) of 1.15 [26]. Conversely, fish oil supplementation showed no association with fecundability [28]. On the other hand, iron supplementation had inconsistent findings on fecundability, with no association found in the Danish cohort, whereas there is a positive FR of 1.19 in the North American cohort [27], suggesting the presence of a population-specific effect. This raised the question of the extent to which those findings can be applied to preconception Asian women, given that the types/forms of supplements consumed, baseline health status, and lifestyle characteristics are varied between populations. Importantly, there is a lack of comprehensive assessment of micronutrient intake commonly consumed from oral supplements, on the TTP of preconception women.

Compared to fertility treatments, dietary supplements are non-invasive and are a more accessible option for couples trying to conceive. In this study, we used data from the Singapore PREconception Study of long-Term maternal and child Outcomes (S-PRESTO) to examine the associations of various types of oral micronutrient supplements, that are commonly consumed during preconception, with fecundability, as measured by TTP, among reproductive-aged Asian women in the general population who were trying to conceive. These findings could pave the way for intervention studies and public health policies to guide the use of preconception supplements to shorten TTP and improve fertility rates.

## 2. Materials and Methods

### 2.1. Study Design and Participants

Data were drawn from the S-PRESTO (ClinicalTrials.gov, NCT03531658), a prospective observational preconception cohort study of 1032 women, examining the long-term effects of exposures before and during early pregnancy on mother-offspring metabolic and mental health outcomes in later life [29]. Asian women of Chinese, Malay, or Indian ethnicity, between 18 and 45 years old, who were trying to conceive within the next year were recruited from the general population of Singapore between February 2015 and October 2017. Women diagnosed with type 1 or type 2 diabetes mellitus, who had attempted to conceive for more than 18 months prior to enrolment, who had taken anticonvulsant medication, oral steroids, or received assisted fertility treatment in the past month were excluded. The study was conducted according to the guidelines laid down in the Declaration of Helsinki. Ethical approval was obtained from the SingHealth Centralised Institute Review Board (reference 2014/692/D). All participants provided their written informed consent.

### 2.2. Study Procedure

At the baseline visit, sociodemographic characteristics, dietary intakes, lifestyle habits, menstrual characteristics, and obstetric histories were obtained by questionnaires through face-to-face interviews, along with anthropometric measurements, conducted by the research staff in the S-PRESTO research centre at the KK Women’s and Children’s Hospital. At the end of the visit, the women received home urinary pregnancy test kits (Biotron Diagnostics, Hemet, CA, USA) and were instructed to perform the test whenever their menstrual period was late three to four days or two weeks after unprotected intercourse. The urinary pregnancy test kit detects the beta subunit of human chorionic gonadotropin at a limit of 25 mIU/mL. If a positive result was detected, women were asked to contact the research staff to schedule an ultrasound scan to confirm the clinical pregnancy. Those with no updates were contacted at 6-, 9-, and 12-month follow-ups by the research staff via telephone to track their pregnancy status. All women were followed for up to a year after recruitment.

### 2.3. Supplement Intake

At the baseline visit, researchers asked women if they had taken any supplements in the past three months and, if so, what specific type of nutrients were in the supplement. Single or multi-supplements containing FA, fish oil, evening primrose oil, iron, zinc, selenium, iodine, and vitamins B6, B12, C, D or E were selected for analysis based on their reported associations with reproductive outcomes [15,16,20,21,22,23,24,25,26,27,28,30,31,32,33]. For each nutrient supplement, women were classified into users or non-users. FA supplement, which is commercially available and commonly consumed in either single or multivitamin form, was classified as a single vitamin, a multivitamin or both according to the responses provided. Peripheral blood was collected from participants after an overnight fast and serum and EDTA plasmas were obtained via centrifugation at 1600 g for 10 min at 4 °C, then aliquoted and stored at −80°C until analysis. Serum vitamin B12 and folate were measured using chemiluminescent immunoassays on the Beckman Access^®^2 immunoassay analyser (Beckman Coulter) in a College of American Pathologists-accredited clinical laboratory. Plasma pyridoxal-5′-phosphate (vitamin B6), 25-hydroxy vitamin D3, and alpha-tocopherol (vitamin E) were measured using targeted liquid chromatography-tandem mass spectrometry assays (BEVITAL AS) [34,35].

### 2.4. Assessment of Time to Pregnancy

Pregnancy was first detected by a positive urinary pregnancy test kit and confirmed by ultrasound of an intrauterine gestational sac after 6 weeks of gestation. TTP was defined as the number of menstrual cycles needed to achieve pregnancy over the one-year follow-up period. The interval between the dates of the last menstrual period (LMP) at baseline and before pregnancy (for pregnant women) or the last follow-up call (for censored women) was calculated and divided by the average cycle length for conversion to cycles. The average cycle length was derived from the reported minimum and maximum cycle lengths. An irregular cycle was defined as having a cycle length that varied by more than five days in the past six months. Data on women’s attempt time to conceive at study entry were also obtained in months. TTP was calculated as the total discrete cycles at risk of pregnancy: (months of conception attempt at study entry/average cycle length) + [(LMP date before conception or last follow-up call) − (LMP date at baseline)]/average cycle length. An additional conception cycle was added for pregnant women [36].

### 2.5. Assessment of Confounders

Confounders were selected via a directed acyclic graph, clinical judgement and a priori from the literature [8,36,37,38,39,40]. They were age (<35, ≥35 years), ethnicity [Chinese, Malay, Indian, Mixed ethnicity (any combination of these groups)], highest education (below tertiary, tertiary and above), parity (0, ≥1), BMI [underweight ( <18.5 kg/m^2^), normal (18.5–22.9 kg/m^2^), overweight (23–27.4 kg/m^2^), obese ( ≥27.5 kg/m^2^)], cycle regularity (regular, irregular), smoking exposure (no, yes), alcohol intake (no, yes), unhealthful plant-based diet index (uPDI) and total daily energy intake. BMI was categorised using cutoffs for Asian populations [41]. Smoking exposure included both active and passive smoking. Alcohol intake was assessed as any alcohol intake in the last three months. Derivation of uPDI and total daily energy intake were based on a validated food frequency questionnaire, as described previously [42]. uPDI was grouped into tertiles, with a higher score denoting consumption of more unhealthful plant-based food components.

### 2.6. Statistical Analysis

The characteristics of women with different supplement intake statuses were compared using Pearson’s chi-square tests for categorical variables and Mann–Whitney tests for continuous variables. Fecundability was assessed using discrete-time proportional hazards modelling with TTP as discrete cycles to calculate the hazard ratio of fecundability, termed FR, and its 95% confidence interval (CI) [36,43]. FR represents the cycle-specific probability of conceiving among exposed women, relative to unexposed women. Unlike the hazard ratio for mortality or morbidity where a ratio of <1.0 is desirable, an FR > 1.0 indicates that exposed women were more likely to conceive at each cycle or shortened TTP. Conversely, an FR < 1.0 indicates that exposed women were less likely to conceive at each cycle or lengthened TTP. Left truncation was accounted for by using only cycles of attempt observed in the study. For example, if a woman conceived after three cycles from baseline and had four cycles of attempt at study entry, only the observed three cycles (fifth to seventh cycles) were used for analysis. Right censoring was performed on women who did not conceive within 12 months of follow-up, initiated fertility treatment, and reported that they were no longer trying or lost to follow-up, whichever occurred first. Both crude models and those adjusted for confounders were analysed.

Given that couples trying to conceive generally seek fertility treatment after 12 months, the fecundability analysis was repeated, restricting the sample to women with an attempt time of three, six, and 12 months at study entry. This helps to control for the variable attempt time at study entry and excludes possible cases of male and female infertility with underlying pathologies. Sensitivity analysis was also performed by excluding cases of a self-reported polycystic ovarian syndrome (PCOS) (*n* = 10) and using singleton live birth as an outcome measure (*n* = 351). Serum/plasma micronutrient levels were categorised into tertiles and their associations with fecundability were analysed. All statistical analyses were performed with SPSS Statistics Version 20 (IBM Corp, Armonk, NY, USA) and Stata Statistical Software, Release 13 (StataCorp, College Station, TX, USA).

## 3. Results

### 3.1. Participant Characteristics

Among the 1032 eligible women, 124 were excluded due to missing or implausible data, resulting in 908 women included in the present study (Figure 1). We defined menstrual cycle data as implausible if there were mismatched information between data collection dates, reported dates of the LMP, cycle length and cycle regularity, which were unable to resolve. Baseline characteristics were similar between included (*n* = 908) and excluded women (*n* = 124) except for cycle regularity, where excluded women were more likely to have irregular cycles (47.7% vs. 34.3%) (Appendix A). The included women contributed 8307 cycles and 387 pregnancies within one year of follow-up. Of the included women, 42.6% (*n* = 387) spontaneously conceived and among them, 268 (29.5%) and 369 (40.6%) women conceived within six and 12 cycles of follow-up, respectively. Attempt time to conceive at study entry for included women was at a median of 1.0 cycle (25th–75th percentile, 0–7.0). Fifty-seven percent (*n* = 521) of the women were censored due to the initiation of fertility treatment (*n* = 14), lost to follow-up (*n* = 19) and did not become pregnant one year after recruitment (*n* = 488). The baseline characteristics of the women are presented in Table 1. Among the 908 included women, 613 (67.5%) had taken some form of supplements in the past three months before recruitment (preconception period); they were more likely to be 35 years of age or older (16.6% vs. 10.8%), of Chinese ethnicity (78.8% vs. 58.0%), nulliparous (70.8% vs. 54.9%), had attained tertiary education and above (68.4% vs. 51.5%), normal BMI (50.2% vs. 38.0%), had no smoking exposure (80.3% vs. 69.8%), had consumed alcohol in the past three months (52.2% vs. 36.9%), and lower total daily energy intake (1918 vs. 2020 kcal) compared to women who were not on any supplements. Cycle regularity and cycle length were similar between women with and without supplement intake.

### 3.2. Associations between Supplement Intake Status and Fecundability

Table 2 shows the associations between the different types of supplement users and fecundability. In model 1, women who took any supplement had significantly higher fecundability compared to women who did not take any supplement (FR 1.34, 95% CI 1.06–1.68). Specifically, FA (1.27, 1.03–1.56) and iodine (1.30, 1.01–1.67) users had significantly higher fecundability relative to non-users. This resulted in a significantly shorter TTP for women taking any supplement (8.80 vs. 9.85 cycles, *p* = 0.001), FA (8.70 vs. 9.61 cycles, *p* = 0.004) and iodine (8.25 vs. 9.33 cycles, *p* = 0.010), compared with respective counterparts (Appendix A). Women who took FA in the form of single vitamins and multivitamins were found to have similar FR. In contrast, evening primrose oil (0.55, 0.31–0.98) was the only supplement that showed significantly lower fecundability among users, relative to non-users. However, the increase in TTP for evening primrose oil users compared with non-users was not significant (9.86 vs. 9.11 cycles, *p* = 0.316) (Appendix A). In model 2, with additional adjustments for uPDI and total daily energy intake, observed findings remained similar.

Similar findings were observed when sensitivity analyses were performed on subsamples of women restricted to those who had attempted conception for ≤3 (*n* = 527), ≤6 (*n* = 653) or ≤12 months (*n* = 774) prior to study entry (Appendix A). Findings were also similar with sensitivity analyses excluding women with PCOS and when singleton live birth was the outcome of interest instead of clinical pregnancy at six weeks (Appendix A).

To corroborate the compliance to supplement use with the self-reported supplement use, we compared available serum/plasma micronutrient levels (FA, vitamins B6, B12, D3, and E) between supplement users and non-users. Measurement of circulating micronutrient levels revealed that users of FA, vitamin B6, B12, D, and E supplements had higher serum micronutrient levels (Appendix A) compared to non-users. Analysis of serum/plasma micronutrient levels on fecundability further showed that women with serum FA > 44.6 nmol/L had higher fecundability than those ≤20.0 nmol/L (1.47, 1.12–1.93) (Appendix A).

## 4. Discussion

In this prospective preconception cohort study that included 908 community-recruited women, we examined the associations between micronutrient supplement intake and fecundability. In general, preconception micronutrient supplementation was associated with better fecundability and shorter TTP of approximately one menstrual cycle in women planning a pregnancy, specifically FA and iodine supplements. Evening primrose oil, on the other hand, was associated with poorer fecundability and slightly longer TTP.

A recent prospective cohort study of 3895 Danish women found that preconception FA supplementation, in addition to reducing the risk of neural tube defects, was associated with higher fecundability when compared to non-users [26]. This is in agreement with our study, which showed a shortened TTP in women taking preconception FA. This extended to any form of FA intake since both single and multivitamins showed a similar magnitude of association with fecundability. The underlying mechanism of FA in improving fertility could be attributed to its involvement in reducing homocysteine levels in the body [44,45,46]. High levels of homocysteine lead to increased oocyte maturity, endothelial inflammation, trophoblast apoptosis, and oxidative stress [9,44,47]. In addition to FA, iodine supplementation was also associated with higher fecundability in our study. A population-based prospective cohort study of 501 women showed that women with lower urinary iodide levels were associated with lower fecundability and took longer to conceive relative to those with sufficient urinary iodide levels [48]. The results of animal studies suggest that iodine, which is an essential component of thyroid hormones, can indirectly improve fertility by regulating ovulation and promoting oocyte fertilisation and embryo implantation [49]. Maternal iodine is also known to play an important role in the neurological development of the foetus [49].

Although the magnitude of improvement of FA and iodine supplements on fecundability as reported here may not be strong enough to influence any change in practice, it may contribute to a greater impact on fecundability when combined with other lifestyle modifications such as smoking, physical activity and alcohol intake [8,37,39], suggesting the need for a holistic approach to improve female fertility. It is also worth noting that vitamin D and E users had only slightly elevated plasma micronutrient levels that may have led to no observed associations with fecundability. Thus, further studies on vitamin D and E supplementation and fecundability are warranted. Nonetheless, the use of these supplements may still be beneficial considering other health benefits including the prevention of cardiovascular disease [50,51].

Evening primrose oil, a form of plant phytoestrogen that resembles vertebrate steroid oestrogen [52,53,54], is commonly used as an alternative therapy for a multitude of ailments [55]. However, many studies on the clinical implications of evening primrose oil are preliminary and have notable methodological flaws [52]. Despite the scarcity of clinical evidence, the evening primrose oil supplement is widely marketed as a product to promote fertility [56]. The hypothesis is that the essential fatty acids it contains improve the quantity and quality of cervical mucus [56,57], which is imperative for the survival and motility of spermatozoa [58]. Despite more reported studies on the impact of phytoestrogens and fertility, the results remain controversial. The analyses of North American (*n* = 4880) and Danish (*n* = 2898) women showed no significant association between pregnancy rates and dietary phytoestrogen intake [24]. However, the Adventist Health Study, which is a cross-sectional study of 11,688 Adventist women in North America aged 30–50 years, showed a 13% higher risk of nulligravidity in women who consumed large amounts (≥40 mg/day) of isoflavones, a soy phytoestrogen [23]. Animal studies also suggested the possible detrimental effects of phytoestrogen on fertility [59]. This is consistent with our finding that women who took preconception evening primrose oil supplementation had 45% lower fecundability. However, given the small proportion of evening primrose oil users in this study, the magnitude of the association may have been exaggerated and careful interpretation is required. Therefore, more research is imperative to determine the impact of evening primrose oil supplementation on fertility.

Our findings suggest that women who are trying to conceive could potentially benefit from taking multivitamin supplements containing FA and iodine. However, the exact dosage remains to be elucidated. As compared to taking multiple pills of different single vitamins, multivitamins may be a convenient option for these women, thereby improving the level of compliance to preconception supplementation [60]. From a public health perspective, supplements are widely accessible to the general population. Leveraging on the global trend of increasing health consciousness and willingness to consume micronutrient supplements, the recommendation of appropriate preconception supplements might have a profound impact on fertility rates worldwide. However, the emphasis in preconception nutrition should be on a healthy and well-balanced diet, with micronutrient supplementation acting as an adjunct to optimise micronutrient reserve and improve fecundability [61].

To our knowledge, this is the only multi-ethnic Asian cohort that comprehensively evaluated various types of micronutrient supplements, including evening primrose oil, on fecundability in women planning pregnancy. We acknowledge the limitations of this study. The main limitation is recall bias, where participants might erroneously report supplement use, resulting in under or overestimation of the association between supplement consumption and fertility. However, the reliability of the data was supported by the higher serum/plasma micronutrient levels of FA, vitamins B6, B12, D3 and E of self-reported users of these supplements, respectively, compared to non-users. Another limitation is behaviour change bias [43], as women with a longer attempt time at study entry may have optimised their lifestyles toward conceiving. However, similar findings were observed when comparing the FR of women trying to conceive for different periods of time (three, six and 12 months) at the start of the study. In addition, male factors that influence fertility were not considered. However, the overall pregnancy rates observed in this study are consistent with the low fertility rate in Singapore and are similar to the pregnancy rate in China [62]. Although our study observed a lower fecundability among evening primrose oil users, the proportion of evening primrose oil users is the lowest, at 5%, which may have exaggerated the magnitude of association. Participants with gynaecological problems that can affect fecundability were not excluded, which may have led to an underestimation of the association between supplement intake and fecundability. Lastly, the impact of micronutrients on fecundability can be dose- and frequency-dependent, however, the concentration and compliance to consumption of the micronutrient supplements were not taken into account.

## 5. Conclusions

Our study showed that dietary intake of preconception micronutrient supplementation, specifically FA and iodine, was associated with higher fecundability or shortened TTP; while evening primrose oil use was associated with lower fecundability or longer TTP. Nonetheless, the magnitude of the association between supplement use and fecundability or TTP will need to be further confirmed with well-designed randomised controlled trials employing standardised doses of FA, iodine and evening primrose oil for treatment groups, along with compliance tracking.

## Figures and Tables

**Figure 1 nutrients-14-05110-f001:**
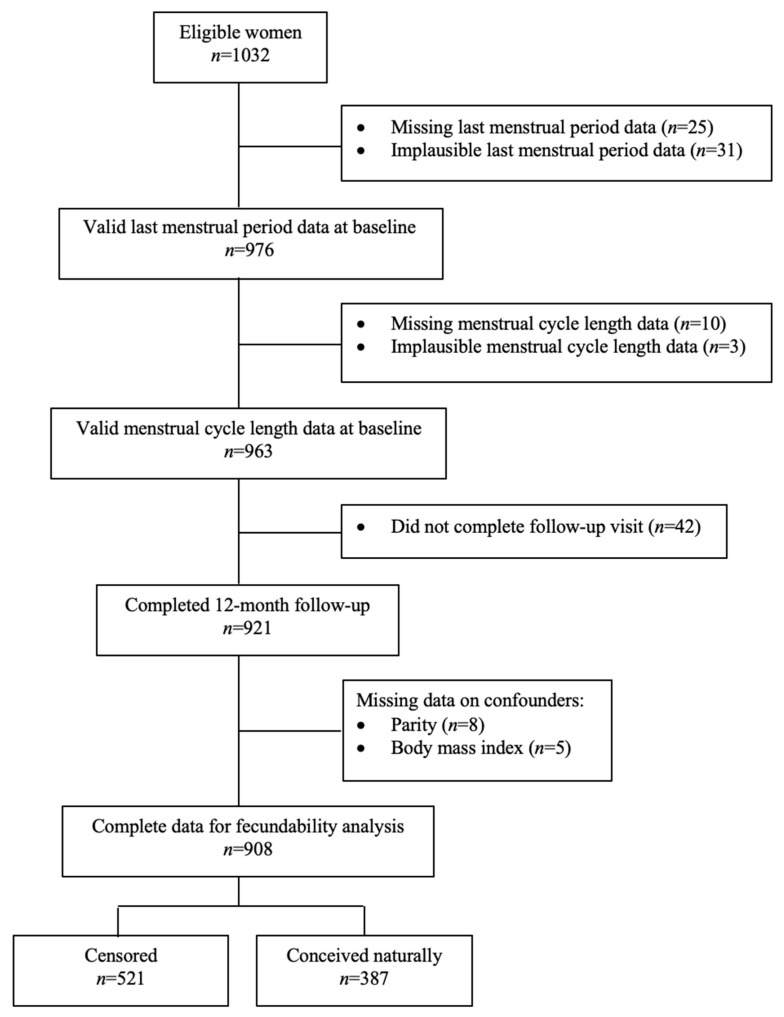
Flowchart showing women included in the present study.

**Table 1 nutrients-14-05110-t001:** Characteristics of women according to supplement intake status in the last 3 months from the S-PRESTO study, 2015–2018 (*n* = 908).

Characteristics	Total (*n* = 908)	No Supplements (*n* = 295)	Any Supplement (*n* = 613)	*p* ^a^
Age				0.021
<35 years	774 (85.2)	263 (89.2)	511 (83.4)	
≥35 years	134 (14.8)	32 (10.8)	102 (16.6)	
Ethnicity				<0.001
Chinese	654 (72.0)	171 (58.0)	483 (78.8)	
Malay	143 (15.7)	75 (25.4)	68 (11.1)	
Indian	82 (9.0)	38 (12.9)	44 (7.2)	
Mix	29 (3.2)	11 (3.7)	18 (2.9)	
Parity				<0.001
0	596 (65.6)	162 (54.9)	434 (70.8)	
≥1	312 (34.4)	133 (45.1)	179 (29.2)	
Highest education				<0.001
Below tertiary	337 (37.1)	143 (48.5)	194 (31.6)	
Tertiary and above	571 (62.9)	152 (51.5)	419 (68.4)	
Body mass index ^b^				<0.001
Underweight <18.5 kg/m^2^	76 (8.4)	21 (7.1)	55 (9.0)	
Normal 18.5–22.9 kg/m^2^	420 (46.3)	112 (38.0)	308 (50.2)	
Overweight 23–27.4 kg/m^2^	238 (26.2)	86 (29.2)	152 (24.8)	
Obese ≥27.5 kg/m^2^	174 (19.2)	76 (25.8)	98 (16.0)	
Cycle regularity				0.053
Regular	597 (65.7)	181 (61.4)	416 (67.9)	
Irregular	311 (34.3)	114 (38.6)	197 (32.1)	
Cycle length, days	29.5 (29.0–32.5)	29.5 (29.0–32.5)	29.5 (29.0–32.5)	0.678
Smoking exposure				<0.001
No	698 (76.9)	206 (69.8)	492 (80.3)	
Yes	210 (23.1)	89 (30.2)	121 (19.7)	
Alcohol intake				<0.001
No	479 (52.8)	186 (63.1)	293 (47.8)	
Yes	429 (47.2)	109 (36.9)	320 (52.2)	
Unhealthful plant-based diet index				0.093
Tertile 1 ≤41	331 (36.5)	95 (32.2)	236 (38.5)	
Tertile 2 >41 to ≤47	290 (31.9)	94 (31.9)	196 (32.0)	
Tertile 3 >47	287 (31.6)	106 (35.9)	181 (29.5)	
Total daily energy intake, kcal/d	1945 (1567–2385)	2020 (1572–2502)	1918 (1553–2317)	0.032
Attempt time to conceive at study entry, cycles	1.0 (0–7.0)	0 (0–6.0)	2.0 (0–8.0)	0.002

Data are presented as number (percentage) for categorical variables and median (25th–75th percentile) for continuous variables. S-PRESTO, Singapore PREconception Study of long-Term maternal and child Outcomes. ^a^ Based on Pearson’s chi-square tests for categorical variables and Mann–Whitney tests for continuous variables. ^b^ Classified according to cutoffs for Asian populations [41].

**Table 2 nutrients-14-05110-t002:** Associations between supplement intake status and fecundability in women from the S-PRESTO study, 2015–2018 (*n* = 908).

				Unadjusted		Model 1 ^a^		Model 2 ^b^	
Type of Supplements	n	Pregnancies	Cycles	FR	95% CI	FR	95% CI	FR	95% CI
Supplement intake status									
No supplements	295	114	698	1.00	(Ref.)	1.00	(Ref.)	1.00	(Ref.)
Any supplement	613	273	1303	1.31	1.05, 1.62	1.34	1.06, 1.68	1.30	1.03, 1.63
Folic acid									
Non-user	441	182	1077	1.00	(Ref.)	1.00	(Ref.)	1.00	(Ref.)
User	467	205	924	1.22	1.00, 1.49	1.27	1.03, 1.56	1.26	1.03, 1.56
Folic acid type									
Non-user	441	182	1077	1.00	(Ref.)	1.00	(Ref.)	1.00	(Ref.)
Single vitamin	235	96	422	1.18	0.92, 1.51	1.25	0.97, 1.61	1.25	0.97, 1.62
Multivitamin	168	79	365	1.23	0.95, 1.61	1.25	0.95, 1.64	1.23	0.94, 1.62
Single vitamin and multivitamin	64	30	137	1.34	0.91, 1.98	1.38	0.93, 2.05	1.39	0.94, 2.07
Fish oil									
Non-user	723	308	1611	1.00	(Ref.)	1.00	(Ref.)	1.00	(Ref.)
User	185	79	390	0.99	0.78, 1.27	0.99	0.77, 1.27	0.98	0.76, 1.26
Evening primrose oil									
Non-user	865	375	1943	1.00	(Ref.)	1.00	(Ref.)	1.00	(Ref.)
User	43	12	58	0.59	0.33, 1.04	0.55	0.31, 0.98	0.56	0.31, 0.99
Iron									
Non-user	688	284	1552	1.00	(Ref.)	1.00	(Ref.)	1.00	(Ref.)
User	220	103	449	1.20	0.96, 1.50	1.20	0.95, 1.51	1.19	0.94, 1.49
Zinc									
Non-user	702	293	1569	1.00	(Ref.)	1.00	(Ref.)	1.00	(Ref.)
User	206	94	432	1.16	0.92, 1.46	1.15	0.91, 1.47	1.13	0.89, 1.43
Selenium									
Non-user	737	308	1634	1.00	(Ref.)	1.00	(Ref.)	1.00	(Ref.)
User	171	79	367	1.17	0.92, 1.50	1.18	0.92, 1.52	1.17	0.91, 1.51
Iodine									
Non-user	749	309	1641	1.00	(Ref.)	1.00	(Ref.)	1.00	(Ref.)
User	159	78	360	1.27	0.99, 1.63	1.30	1.01, 1.67	1.28	1.00, 1.65
Vitamin B6									
Non-user	676	278	1487	1.00	(Ref.)	1.00	(Ref.)	1.00	(Ref.)
User	232	109	514	1.19	0.96, 1.49	1.18	0.94, 1.48	1.17	0.93, 1.47
Vitamin B12									
Non-user	673	276	1491	1.00	(Ref.)	1.00	(Ref.)	1.00	(Ref.)
User	235	111	510	1.21	0.97, 1.51	1.20	0.96, 1.50	1.19	0.95, 1.49
Vitamin C									
Non-user	615	245	1337	1.00	(Ref.)	1.00	(Ref.)	1.00	(Ref.)
User	293	142	664	1.27	1.03, 1.56	1.22	0.98, 1.51	1.19	0.96, 1.47
Vitamin D									
Non-user	687	285	1499	1.00	(Ref.)	1.00	(Ref.)	1.00	(Ref.)
User	221	102	502	1.14	0.91, 1.43	1.18	0.94, 1.48	1.15	0.91, 1.45
Vitamin E									
Non-user	704	297	1591	1.00	(Ref.)	1.00	(Ref.)	1.00	(Ref.)
User	204	90	410	1.08	0.85, 1.37	1.11	0.88, 1.42	1.09	0.86, 1.39

Data were analysed using the discrete-time proportional hazards model. S-PRESTO, Singapore PREconception Study of long-Term maternal and child Outcomes; FR, fecundability ratio; CI, confidence interval; Ref., reference. ^a^ Model 1: adjusted for age, ethnicity, education, parity, body mass index, cycle regularity, smoking exposure, and alcohol intake. ^b^ Model 2: Model 1 + unhealthful plant-based diet index and total daily energy intake.

## Data Availability

Please contact the corresponding author for more information.

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
