# Peer review of "Dietary Supplement Intake and Fecundability in a Singapore Preconception Cohort Study"

_nutrients, 2022, doi:10.3390/nu14235110_

Round 1
Reviewer 1 Report
The manuscript "Dietary supplement intake and fecundability in a Singapore preconception cohort study" is an interesting manuscript on the correlation between dietary supplement intake and fecundability. The work is not so original, but it is well performed and structured. The design of the project is appropriate and the results are significant. The statistical analysis is well conducted and the language is acceptable. The main concern of this work is related to the clinical condition of these patients: do the authors consider these women affected by any gynaecological problem (tubal factor, endometriosis, or any specific infertility cause)? This could be an important bias of the work, so I'll suggest adding it in the limitation paragraph.
There are some references in the reference list which are too old, so I suggest removing them (Hirsch, A.M.; Hirsch, S.M. The effect of infertility on marriage and self-concept. J Obstet Gynecol Neonatal Nurs 1989, 18, 13-20, doi:10.1111/j.1552-6909.1989.tb01611.x.; Spira, A. The use of fecundability in epidemiological surveys. Hum Reprod 1998, 13, 1753-1756, doi:10.1093/oxfordjournals.humrep.a019710; Hearnshaw, H.; Brown, J.M.; Cumming, I.A.; Goding, J.R.; Nairn, M. Endocrinological and histopathological aspects of the infertility in the ewe caused by oetrogenic clover. J Reprod Fertil 1972, 28, 160-161, doi:10.1530/jrf.0.0280160.; Saloniemi, H.; Wähälä, K.; Nykänen-Kurki, P.; Kallela, K.; Saastamoinen, I. Phytoestrogen content and estrogenic effect of 525 legume fodder. Proc Soc Exp Biol Med 1995, 208, 13-17, doi:10.3181/00379727-208-43825).
It represents an interesting work and it gives the opportunity to focus the attention on the correlation between dietary supplement intake and fecundability.
Author Response
Please see the attachment for the reply to Reviewer 1

Reviewer 2 Report
This paper investigates supplement intake and time to pregnancy. The sample is from the PRESTO cohort, allowing the authors to utilise a large sample size to investigate these associations.
Introduction. The introduction provides some rationale for why supplement intakes may relate to fecundability but this needs to be much stronger. What components in these supplements are important, and why are they better than the many diet studies that have now looked at fecundability? This needs to be emphasised to provide any sort of novelty or update to current literature. Some additional quantitative information from previous studies would also be helpful, along with why the choice to analyse many supplements rather than just a select few. The first paragraph of the introduction would benefit from revision as it is repetitive and has a number of grammatical issues.
Methods: what is the sample size of the cohort? This could be stated in the first paragraph.
The statistical analyses and confounders used are appropriate. Why was dietary intake not included as a confounder? Food intake is often related to supplement intake but also fecundity.
Results. The results were well written
I don’t understand however why women who started initiation of fertility treatment were censored. Could there not be a sensitivity analysis including only those who conceived without treatment?
How was implausible menstrual cycle data defined?
In the supplementary analysis on compliance to supplementation, it would be useful to see how these concentrations related to time to pregnancy. This would strengthen the indication the supplement users were in fact compliant and that plasma concentrations also relate to fecundity. Even though the plasma values for Vitamin D and E show significant differences between groups, these are actually quite small and I would not assume this is the result of supplemental intake. Hence, another reason to analyse TTP using plasma levels.
Can you quantify the difference in median time to pregnancy between groups, for example see the paper by Grieger et al, 2018. Human Reproduction. It is difficult to interpret fecundity in percentages.
Discussion.
The discussion would benefit from a greater synthesis of results. I understand that the authors are reporting on different supplements and the results, but there is not a comprehensive synthesis of findings. I imagine this is likely related to the fact that few studies have assessed supplements and fecundity but this goes back to the introduction which additionally requires a stronger rationale. It is important to get the point across as to why the study was undertaken and what previous work has hinted at potential relationships, and so in the discussion, the study findings can be discussed in relation to these. The paragraph on EPO is ok but some grammatical issues (and throughout the discussion) need to be amended, and that animal studies usually come before human studies so these don’t necessarily “echo” the studies in humans, as it currently states. Further explanation on why the percentages mean in terms of fecundity is also needed. Is 27% and 30% a lot? What would be the equivalent in months? For readers with less understanding of fecundity, it would be helpful to highlight whether higher or lower fecundity is ‘better’.
The paragraph on public health implications and preconception health is not strong, and given few studies have reported on supplements and fecundity, these will not suffice to change practice or promote supplements to improve time to pregnancy. What doses of supplements were they taking? Some comment on the plasma concentrations and small differences in vit D and E should be elaborated on.
The conclusion is a bit overstated and requires interpretation in terms of whether the percentage differences in fecundity is meaningful.
Other comments:
The reference section should be checked as it includes many websites, and a number of early studies where there are more recent ones available.
Please remove abbreviations such as EPO, as these are not common and make reading difficult.
Author Response
Please see the attachment for the reply to Reviewer 2
